# A High-Frequency Vibration Error Compensation Method for Terahertz SAR Imaging Based on Short-Time Fourier Transform

Yinwei Li [1], Qi Wu [2], Jiawei Jiang [2], Xia Ding [3], Qibin Zheng [1] and Yiming Zhu [1,*]

1. Terahertz Technology Innovation Research Institute, Terahertz Spectrum and Imaging Technology Cooperative Innovation Center, Shanghai Key Lab of Modern Optical System, University of Shanghai for Science and Technology, Shanghai 200093, China; liyw@usst.edu.cn (Y.L.); qbzheng@usst.edu.cn (Q.Z.)
2. School of Optoelectronic Information and Computer Engineering, University of Shanghai for Science and Technology, Shanghai 200093, China; 192380289@st.usst.edu.cn (Q.W.); 202310315@st.usst.edu.cn (J.J.)
3. Shanghai Radio Equipment Research Institute, Shanghai 201109, China; miragedx@sina.com
* Correspondence: ymzhu@usst.edu.cn

**Abstract:** High-frequency vibration error of a moving radar platform easily introduces a non-negligible phase of periodic modulation in radar echoes and greatly degrades terahertz synthetic aperture radar (THz-SAR) image quality. For solving the problem of THz-SAR image-quality degradation, the paper proposes a multi-component high-frequency vibration error estimation and compensation approach based on the short-time Fourier transform (STFT). To improve the robustness of the method against noise effects, STFT is used to extract the instantaneous frequency (IF) of a high-frequency vibration error signal, and the vibration parameters are coarsely obtained by the least square (LS) method. To reduce the influence of the STFT window widths, a method based on the maximum likelihood function (MLF) is developed for determining the optimal window width by a one-dimensional search of the window widths. In the case of high noise, many IF estimation values seriously deviate from the true ones. To avoid the singular values of IF estimation in the LS regression, the random sample consensus (RANSAC) is introduced to improve estimation accuracy. Then, performing the STFT with the optimal window width, the accurate vibration parameters are estimated by LS regression, where the singular values of IF estimation are excluded. Finally, the vibration error is reconstructed to compensate for the non-negligible phase of the platform-induced periodic modulation. The simulation results prove that the error compensation method can meet THz-SAR imaging requirements, even at a low signal-to-noise ratio (SNR).

**Keywords:** high-frequency vibration error; maximum likelihood function (MLF); short-time Fourier transform (STFT); random sample consensus (RANSAC); terahertz synthetic aperture radar (THz-SAR)

## 1. Introduction

Terahertz (THz) waves are the electromagnetic waves between 0.1 THz and 10 THz; they are in the gap between microwave and infrared. In terms of their energy, they are between electrons and photons [1,2]. Therefore, THz waves have many unique advantages that microwaves and infrared do not have. For example, THz waves can generate a higher carrier frequency and larger absolute bandwidth than microwaves. At the same time, THz radiation can penetrate materials such as ceramics, fats, fabrics, plastics, etc., with very little attenuation. In addition, when it penetrates the human body, THz radiation is harmless due to its lower photon energy. These unique characteristics allow THz imaging to be used in many fields [3–7].

The concept of a synthetic aperture imaging technique in the microwave band has been introduced in the THz band [8,9]. Compared to microwave synthetic aperture radar (SAR), THz-SAR can achieve higher two-dimensional image resolution. Additionally, with

its shorter synthetic aperture time, THz-SAR imaging has a higher frame rate [10,11]. Theoretically, the SAR principle requires uniform linear movement of the antenna phase center (APC). In practice, due to the influence of atmospheric turbulence and artificial control, APC always deviates from the ideal motion state, resulting in motion error. According to the variations of the motion error in the whole synthetic aperture, it can be divided into low-frequency motion error and high-frequency vibration error. The low-frequency motion error ranges from the centimeter level to the meter level, but now there are many compensation methods to eliminate its influence on SAR imaging. Meanwhile, the vibration amplitude of high-frequency vibration error is very small, usually at the millimeter level or even the sub-millimeter level, such that its influence can be ignored in microwave SAR imaging. However, since the wavelengths of THz waves are relatively short and comparable to the vibration amplitude of high-frequency vibration error, its influence has to be considered in THz-SAR imaging.

Currently, there exist two kinds of high-frequency vibration-error compensation methods. Similar to motion-error compensation in microwave SAR, phase errors caused by high-frequency vibration error is estimate directly through extended phase gradient autofocus (PGA) [12,13], where the specific vibration information is still ambiguous after phase-error compensation. Therefore, some researchers have focused on another class of methods based on parameter estimation, whose main idea is to estimate the vibration-signal parameters. Sinusoidal frequency modulation Fourier transform (SFMFT) and wavelet multi-resolution analysis are directly utilized to operate on the phase term of the vibration signal through phase unwrapping in [14–16]. However, these methods will fail when the phase ambiguity is serious.

In [17], the discrete fractional Fourier transform (DFrFT) is used to reconstruct the vibration signal with the estimated instantaneous vibration accelerations and vibration frequency. In [18], the vibration parameters are estimated based on the wavelet transform. The adaptive chirplet decomposition and the short-time Fourier transform (STFT) are used in [19] and [20], respectively, and an estimation method considering the variation of vibration amplitude is proposed in [21]. However, the above methods may fail in dealing with multiple frequency component vibration.

The discrete sinusoidal frequency modulation transforms (DSFMT) and the optimization method are combined to estimate the multi-component vibration parameters in [22,23]. However, it inevitably results in large estimation errors because of the convergence to a local minimum. In [24], an improved DFrFT-based estimation method is proposed to estimate multi-component vibration parameters. However, there exists a great correlation between the method accuracy and the window width.

In reality, all the above estimation methods must first extract the vibration signal of the strong point target in the coarsely focused image. The extracted vibration signal will be disturbed by the surrounding clutter signal, and the signal-to-clutter ratio may be fairly low. Although it is biased in estimating instantaneous frequency (IF), STFT is comparatively robust against noise effects [25]. Thus, a novel parameter estimation method of multi-component high-frequency vibration error based on the STFT is presented in this paper. After IF estimates using STFT, least square (LS) regression is performed on the IF estimates to estimate the high-frequency vibration parameters. The estimated parameters are insensitive to the noise effects. However, the method accuracy is influenced by the window width. To overcome the problem, the parameters are estimated for various window widths in the STFT. Then, a method based on the maximum likelihood function (MLF) is developed for determining the optimal window width by a one-dimensional search of these window widths. Here, the multi-dimensional search for a set of parameters is avoided, thus reducing the computing time. In addition, the method accuracy is also influenced by the accuracy of the IF-estimate samples in the LS regression. If the noise is significant, the estimation error is aggravated by the IF estimates that deviate significantly from the true value. To avoid singular IF estimates in the LS regression, a random sample consensus (RANSAC) [26,27] is introduced to randomly select the IF-estimate samples for

multiple times. The previous MLF can also be used as a measure of the selection results. Finally, to show the superiority of the proposed method, it is compared with the original STFT-based method.

The rest of this paper is organized as follows. Section 2 establishes the high-frequency vibration error model and the THz-SAR echo-signal model. The proposed estimation and compensation method is described in detail in Section 3. Section 4 gives the simulation results, and conclusions are drawn in Section 5.

## 2. THz-SAR Echoes Signal Model with Vibration Error

### 2.1. High-Frequency Vibration Error Model of Platform

In SAR imaging, besides the translational motion error, the vibration error of the motion platform is also inevitable because of the effects of air flow and engine vibration. Compared with traditional translational motion error, the motion platform's vibration is of the higher vibration frequency that usually meets the conditions.

$$|f \cdot T_s| \geq 1 \tag{1}$$

where $T_s$ is the synthetic aperture time of SAR.

So far, the vibration of motion platform is usually modeled as a simple harmonic vibration, which is expressed as

$$\Delta R(t) = a \sin(2\pi f t + \phi) \tag{2}$$

where $t$ represents the azimuth time, $a$ denotes the vibration amplitude, $\phi$ represents the initial phase, and $f$ is vibration frequency.

However, in reality, the high-frequency vibration of the motion platform is complex and modeled as a superposition of multiple harmonic vibration. Therefore, to characterize the motion platform vibration more accurately in this paper, we model it with the form of

$$\Delta R(t) = \sum_{i=1}^{I} a_i \sin(2\pi f_i t + \phi_i) \tag{3}$$

where $I$ represents the number of vibration components, $a_i$, $f_i$, and $\phi_i$ denote the vibration amplitude, vibration frequency and initial phase of the $i$'th vibration component.

### 2.2. THz-SAR Echo Signal Model

Figure 1 gives the THz-SAR imaging geometric model when there is high-frequency vibration error. The radar operates in right side-looking strip-map mode. Ideally, the radar platform moves horizontally in the X direction at a reference altitude $H$ and a constant velocity $v$.

The coordinate of the ideal position of the APC is $(vt, 0, H)$. Take a point scatterer, $P$, with the coordinate $(vt_0, y_0, 0)$ as an example, where $t_0$ is the zero doppler time and $y_0$ is the Y-axis position. The ideal instantaneous slant range $r(t)$ from the APC to the point is

$$r(t) = \sqrt{r_0^2 + v^2(t - t_0)^2} = r_0 + \frac{v^2(t - t_0)^2}{2r_0} \tag{4}$$

where $r_0 = \sqrt{y_0^2 + H^2}$ is the nearest slant range of point scatterer.

However, in practical applications, the actual trajectory of the APC deviates from the ideal motion state because of the vibration error and is indicated by the curve line. Thus, the instantaneous slant range is affected by high-frequency vibration error. The actual instantaneous slant range $R(t)$ is

$$R(t) = r(t) + \Delta R(t) \tag{5}$$

Assume that the linear frequency modulation (LFM) pulse signal emitted by THz-SAR is

$$s(\tau) = \text{rect}\left(\frac{\tau}{T}\right) \exp(j2\pi f_0\tau + j\pi\gamma\tau^2) \tag{6}$$

where $\text{rect}(\cdot)$ denotes the rectangular window function, $\tau$ represents the fast time, $T$ denotes the signal pulse width, $f_0$ represents the carrier frequency, and $\gamma$ denotes the chirp rate.

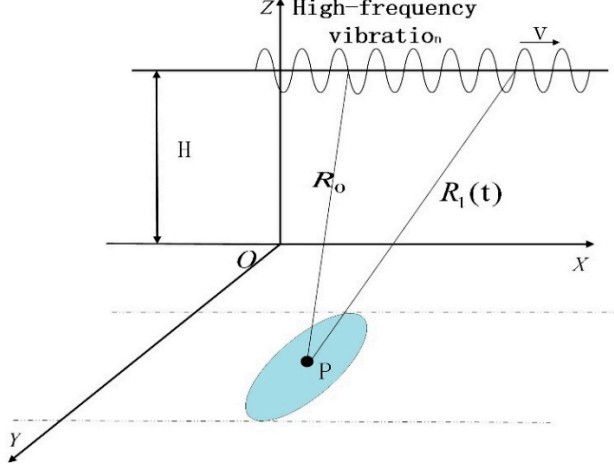

**Figure 1.** THz-SAR imaging geometric model.

In THz-SAR imaging, the radar system usually uses the dechirp heterodyne receiver [28]. After compensating the residual video phase, the recorded baseband echo signal is

$$s(\tau, t) = \sigma \cdot \text{rect}\left[\frac{\tau - 2R(t)/c}{T}\right] \exp\left\{-j\frac{4\pi f_0}{c}R(t) + j\frac{4\pi\gamma R(t)}{c}\tau\right\} \tag{7}$$

where $c$ is the speed of light.

Then, through performing the fast Fourier transform (FFT) upon $\tau$ and a systematic range cell-migration correction (RCMC), the range-compressed signal is

$$s(f, t) = \delta\left(f - \gamma\frac{2r_0}{c}\right)\exp\left\{-j\frac{4\pi}{\lambda}r\right\}\exp\left\{-j\frac{4\pi}{\lambda}\Delta R\right\} \tag{8}$$

where $\delta(\cdot)$ denotes the impulse function and $\lambda$ represents the wavelength.

Substituting (3) and (4) into (8), we can obtain

$$s(f, t) = \delta\left(f - \gamma\frac{2r_0}{c}\right)\quad \exp\left\{-j\frac{4\pi}{\lambda}\left(r_0 + \frac{v^2(t-t_0)^2}{2r_0}\right)\right\}$$
$$\exp\left\{-j\frac{4\pi}{\lambda}\sum_{i=1}^{I}a_i\sin(2\pi f_i t + \phi_i)\right\} \tag{9}$$

Then, after performing the dechirp operation and the residual video phase compensation in azimuth, (9) can be rewritten as

$$s(f, t) = \delta\left(f - \gamma\frac{2r_0}{c}\right)\quad \exp\left\{-j\frac{4\pi}{\lambda}\left(r_0 + \frac{v^2 t_0}{r_0}t\right)\right\}$$
$$\exp\left\{-j\frac{4\pi}{\lambda}\sum_{i=1}^{I}a_i\sin(2\pi f_i t + \phi_i)\right\} \tag{10}$$

Finally, the azimuth compression in the Doppler domain can be accomplished by performing the FFT upon $t$ in (10). However, the high-frequency vibration error brings the additional period-modulated phase into the ideal echo, as shown in the last term of (10). The phase error results in the paired echo in the THz-SAR images.

## 3. Compensation Method

### 3.1. STFT-Based Rough Estimation

First, we select the dominant point target from the coarsely focused image. The signal of interest (SoI) is obtained through performing inverse FFT to convert the coarsely focused image to the range-compressed phase history domain. To eliminate the linear phase in SoI, similar to PGA [29], the dominant point target is circularly shifted to the image center before performing inverse FFT. Thus, after accomplishing the above operations, we obtain the SoI, which is expressed as

$$s(t) = \exp\left\{-j\frac{4\pi}{\lambda}\sum_{i=1}^{I}a_i\sin(2\pi f_i t + \phi_i)\right\} \tag{11}$$

The signal is sampled with the sampling time $\Delta t = 1/f_{PRF}$, where $f_{PRF}$ is the pulse repetition frequency. The discretized SoI signal $s(n)$ is represented as

$$s(n) = s(n\cdot\Delta t), \ n \in [-N/2, N/2] \tag{12}$$

where $N$ represents the sample point number.

On the one hand, the IF can be estimated by using the STFT. That is to say

$$\hat{f}_h(n) = \arg\max_{f}|\text{STFT}_h(n,f)| \tag{13}$$

where $\text{STFT}_h(n, f)$ is the STFT of the signal with window width $h$.

On the other hand, the IF of the signal can be acquired by taking the derivative of the phase in (11):

$$f(t) = \frac{2}{\lambda}\sum_{i=1}^{I}2\pi f_i a_i\cos(2\pi f_i t + \phi_i) \tag{14}$$

After discretization, the IF in (14) can be rewritten as

$$f(n) = \sum_{i=1}^{I}[C_{0,i}\cos(2\pi f_i n\Delta t) + S_{0,i}\sin(2\pi f_i n\Delta t)] \tag{15}$$

where

$$\begin{cases} C_{0,i} = \frac{2}{\lambda}2\pi f_i a_i\cos\varphi_i \\ S_{0,i} = \frac{2}{\lambda}2\pi f_i a_i\sin\varphi_i \end{cases} \tag{16}$$

After estimating the IF $\hat{f}_h(n)$ through (13), the high-frequency vibration parameters are obtained by solving the following linear equations, namely:

$$\hat{\mathbf{f}} = \mathbf{M}\mathbf{x} \tag{17}$$

where

$$\begin{cases} \hat{\mathbf{f}} = \left[\hat{f}(-Q/2), \ \hat{f}(-Q/2+1), \cdots, \hat{f}(Q/2-1), \hat{f}(Q/2)\right]^{\text{T}} \\ \mathbf{x} = [C_{0,1}, S_{0,1}, C_{0,2}, S_{0,2}, \cdots C_{0,I}, S_{0,I}]^{\text{T}} \\ \mathbf{M} = [\mathbf{a}_{0,1}, \mathbf{b}_{0,1}, \mathbf{a}_{0,2}, \mathbf{b}_{0,2}, \cdots \mathbf{a}_{0,I}, \mathbf{b}_{0,I}] \\ \mathbf{a}_{0,i} = [cos(2\pi f_i(-Q/2)\Delta t), \cdots cos(2\pi f_i(Q/2)\Delta t)]^{\text{T}} \\ \mathbf{b}_{0,i} = [sin(2\pi f_i(-Q/2)\Delta t), \cdots sin(2\pi f_i(Q/2)\Delta t)]^{\text{T}} \end{cases} \tag{18}$$

and $Q + 1$ is the number of azimuth instants.

According to (14), the frequencies of the IF are the same as those of the vibration. Thus, through performing the spectral analysis on the estimated IF, the vibration frequencies are obtained. Based on the estimated vibration frequency, the LS solution of (17) is

$$\hat{\mathbf{x}} = (\mathbf{M}^{\text{T}}\mathbf{M})^{-1}\mathbf{M}^{\text{T}}\hat{\mathbf{f}} \tag{19}$$

Thus, the other high-frequency vibration parameters are obtained by

$$a_i = \frac{\sqrt{\hat{\mathbf{x}}^2(2i-1) + \hat{\mathbf{x}}^2(2i)}}{\frac{2}{\lambda}2\pi\hat{f}_i} \tag{20}$$

$$\phi_i = \arctan\left(\frac{\hat{\mathbf{x}}(2i)}{\hat{\mathbf{x}}(2i-1)}\right) \tag{21}$$

Although the obtained parameters are insensitive to noise, the method's accuracy is affected by the window width. It is due to that the STFT has the limitation that it cannot satisfy the optimal frequency resolution and time resolution at the same time [30,31]. The window width $h$ of STFT will bring errors to the estimation results. Meanwhile, the accuracy of the IF estimation samples in the LS regression also influences the method's accuracy, especially in weak targets surrounded by clutter. Owing to these two factors, there may be a large deviation between the estimated values and the actual values, which cannot meet the requirement of THz-SAR vibration error compensation. Thus, THz-SAR images are still influenced by residual phase errors.

### 3.2. Improved STFT-Based Fine Estimation

Owing to the window width, $h$, and the singular values of IF estimation, the estimated vibration parameters values may have a large deviation from the true ones. Here, MLF and RANSAC are introduced to the STFT-based estimation, which can improve the estimation accuracy.

Firstly, the IF is estimated by the STFT for various window widths. The set of various window widths in the STFT is given as

$$H = \{h|h_{min} \leq h \leq h_{max}\} \tag{22}$$

where $h_{min}$ and $h_{max}$ are the minimum and maximum window width, which can be set according to the size of the sampling data. For each window width $h \in H$, vibration parameters are estimated through the above STFT-based estimation method, and a set of vibration parameters $\left\{\hat{a}_i, \hat{f}_i, \hat{\phi}_i\right\}$ can be obtained. Then, to get the optimal window width, one-dimensional search is performed on the window width of STFT. In this paper, the MLF is adopted as the evaluation function.

$$G(h) = \left|\sum_n s(n) \cdot \exp(j\frac{4\pi}{\lambda}\sum_i^I \hat{a}_i \sin(2\pi\hat{f}_i n\Delta t + \hat{\phi}_i))\right| \tag{23}$$

When the value of MLF is maximal, the corresponding window width is the optimal window, and the vibration parameters under the optimal window are to be, themselves, optimal, obtained through

$$\left\{\hat{a}_i, \hat{f}_i, \hat{\phi}_i\right\} = \arg\max_h G(h) \tag{24}$$

In this way, the estimation error caused by window width of STFT is greatly reduced and the parameter estimation accuracy is improved. What is more, the 3I-dimensional search over the parameter space is replaced by a one-dimensional search of window width. Compared with the STFT-based estimation method, the determination of the optimal window only adds a small amount of computation.

Although the refinement of the optimal window width determination can improve the estimation accuracy, the singular values of IF estimation still have a negative influence on the method performance. As the noise level increases, the percentage of the singular values of IF estimation rapidly increases and, with it, the greater the influence on the estimation accuracy. Thus, on the basis of determining optimal window width, RANSAC is used to reduce the estimation error caused by the singular values of IF estimation. A summary of the proposed STFT-based fine estimation method is described next.

Step 1. Initialize a set $H$ of window width, and set the current value of MLF to $G = 0$.

Step 2. For window width $h_j \in H$, the STFT of the signal $s(n)$ is calculated and IF estimates are obtained through (13).

Step 3. Vibration parameters $\left\{ \hat{a}_i, \hat{f}_i, \hat{\phi}_i \right\}$ are estimated by the LS method.

Step 4. On the basis of the estimated vibration parameters, use (23) to calculate the MLF $G(h)$. If $G(h) > G$, set $G = G(h)$, $h = h_j$ and record the corresponding estimated parameters values.

Step 5. Determine whether to traverse the window width completely. If not, repeat Step 2 to Step 4; otherwise, the current window width is optimal.

Step 6. Set the iteration number M, set the parameter values and the criterion function under the optimal window width as the initialization vibration parameters and the initialization $G$, separately.

Step 7. For the optimal window width $h$, use the STFT to estimate the IF through (13).

Step 8. When the vibration parameters are estimated by LS for the $m$'th time, the 2I IF estimation values are randomly selected from the $Q + 1$ IF values estimated in Step 7. The corresponding discrete time of these 2I IF values is $n_m^{(q)}$, $q = 1, 2, \cdots, 2I$ and $n_m^{(q)} < n_m^{(q+1)} < Q$. These selected IF estimation values form the set $\hat{\mathbf{f}}^{(m)}$.

$$\hat{\mathbf{f}}^{(m)} = \left\{ \hat{f}\left(n_m^{(q)} \cdot \Delta t\right), q = 1, 2, \cdots, 2I \right\} \tag{25}$$

Step 9. LS method is used to estimate the vibration parameters, namely

$$\hat{\mathbf{x}}^{(m)} = \left(\mathbf{K}^T \mathbf{K}\right)^{-1} \mathbf{K}^T \hat{\mathbf{f}}^{(m)} \tag{26}$$

where

$$\begin{cases} \mathbf{K} = [\mathbf{c}_{0,1}, \mathbf{d}_{0,1}, \mathbf{c}_{0,2}, \mathbf{d}_{0,2}, \cdots \mathbf{c}_{0,I}, \mathbf{d}_{0,I}] \\ \mathbf{c}_{0,i} = \left[\cos\left(2\pi f_i n_m^{(1)} \Delta t\right), \cdots \cos\left(2\pi f_i n_m^{(2I)} \Delta t\right)\right]^{\mathrm{T}} \\ \mathbf{d}_{0,i} = \left[\sin\left(2\pi f_i n_m^{(1)} \Delta t\right), \cdots \sin\left(2\pi f_i n_m^{(2I)} \Delta t\right)\right]^{\mathrm{T}} \end{cases} \tag{27}$$

Then, use (20) and (21) to calculate the parameters $\{\hat{a}_{i,m}, \hat{\phi}_{i,m}\}$.

Step 10. On the basis of the estimated vibration parameters, use (23) to calculate the MLF $G(m)$. If $G(m) > G$, set $G = G(m)$ and record the corresponding estimated parameters values.

Step 11. If $m < M$, repeat Step 8 to Step 10. Otherwise, the current estimation values are the final estimation results.

After the above improved STFT-based fine estimation, the effects of the STFT window width and the singular values of IF estimation have been reduced as much as possible. The estimate accuracy of vibration parameters has been greatly improved.

### 3.3. Vibration Compensation Method

In THz-SAR imaging, the high-frequency vibration error leads to the paired echo and reduces the image quality when the imaging algorithms are directly performed on the echo data. Here, a multi-component high-frequency vibration error compensation method based on STFT is proposed for solving the above problem.

First, the dominant point target is selected from the coarsely focused image and the SoI is extracted for the vibration-parameters estimation. Then, the accurate vibration error parameters are estimated by the improved STFT-based fine estimation method. Finally, on the basis of the optimal estimated parameters values, the vibration error is compensated, and the THz-SAR focused image is obtained. Therefore, the flow chart of the proposed vibration error compensation method is given in Figure 2.

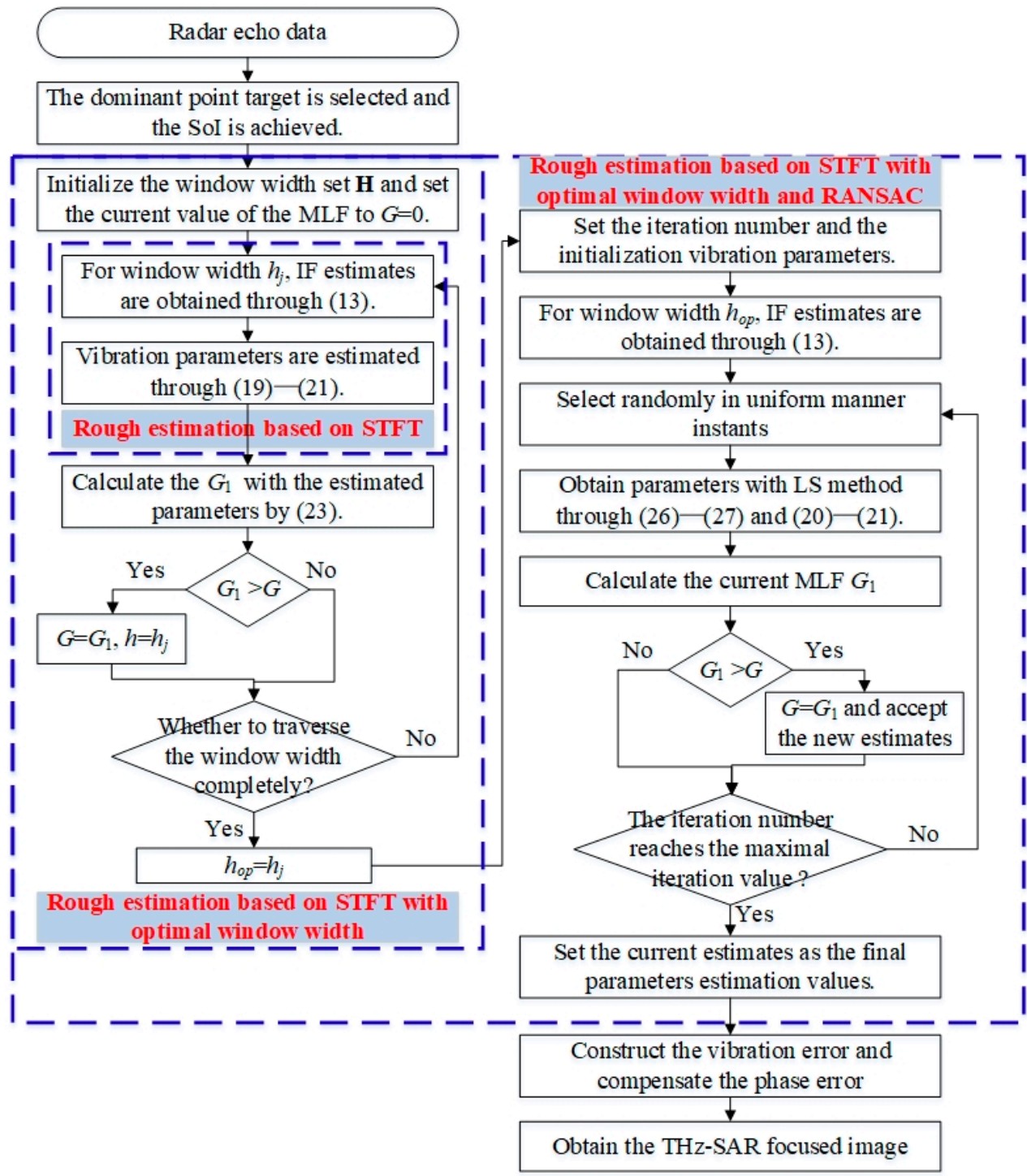

**Figure 2.** Flow chart of the proposed vibration error compensation estimate method.

## 4. Simulation Analysis

In this section, the STFT-based rough estimation, the STFT-based rough estimation with the optimal window width and the improved STFT-based fine estimation are firstly used to estimate the vibration parameters. In a different SNR, the estimation performances of these three methods are compared. Then, through using these methods to estimate and compensate for vibration error in THz-SAR imaging, the effectiveness of the proposed method is further verified.

### 4.1. Estimation Performance Comparison

In this sub-section, simulations are given to evaluate the performance of the proposed parameter estimation method. Firstly, a single-component high-frequency vibration signal is considered and takes the form of (11). Signal parameters are set as $a_1 = 2.00$ mm, $f_1 = 10$ Hz, $\phi_1 = \pi/3$; and, in the noisy scenario, the signal-to-noise ratio (SNR) is defined as SNR $= 1/\sigma^2$, where $\sigma^2$ is the variance of white Gaussian noise.

The SNR of the high-frequency vibration signal changes by 1-dB increments between $-2$ dB and 6 dB. For each SNR value, the above parameters estimation methods are performed to obtain the signal parameters, and one hundred simulations are performed. The normalized root–mean–square error (NRMSE), as shown in [32], is used to evaluate the estimation performance.

$$\text{NRMSE} = \frac{\sqrt{\frac{1}{N}\sum_{i=1}^{N}(\hat{x}_i - x)^2}}{|x|} \tag{28}$$

where $N$ denotes the number of simulations, $\hat{x}_i$ is the estimated value for the i'th simulation, and $x$ is the true value.

For different parameters estimation methods, Figure 3 gives the measured NRMSEs of the high-frequency vibration-signal parameters within different SNR circumstances. Here, in the STFT-based rough estimation method, we chose 20 and 30 as the window width. In the latter two methods, the optimal window width, $h_{op}$, is 26, which is determined by maximizing MLF in (23). The STFT-based rough estimation method produces different estimation accuracies for different window widths. It can be obviously seen from Figure 3 that the improved STFT-based fine estimation has the best performance in rather low SNR and the performance difference decreases as SNR increases. When SNR is higher than 2 dB, all three methods can effectively estimate the SFM signal parameters.

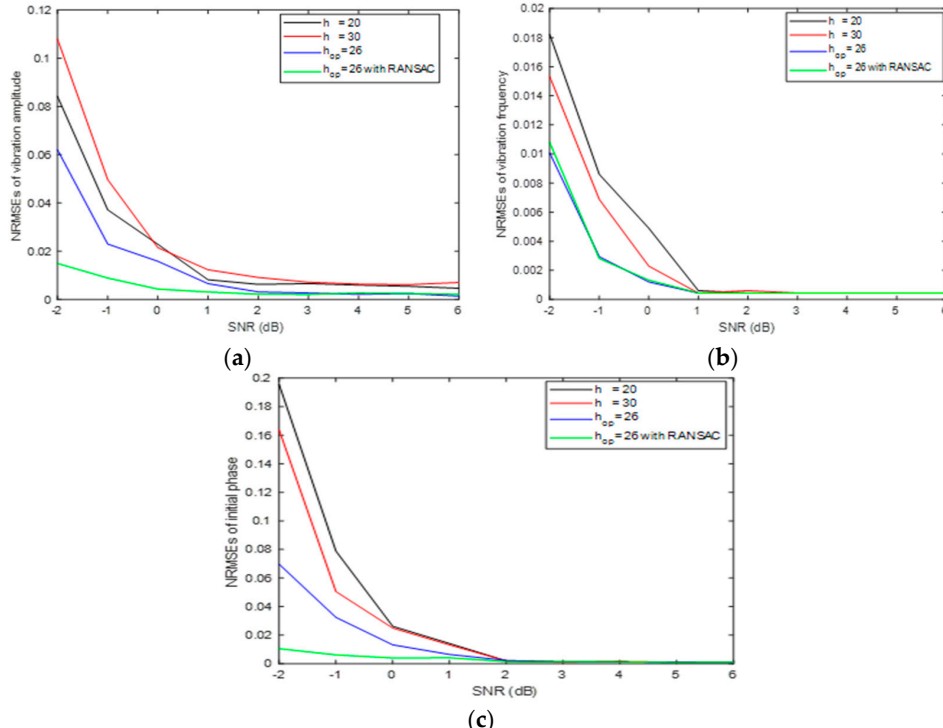

**Figure 3.** The estimation accuracy of signal parameters for different estimation methods. (**a**) vibration amplitude $a_1$. (**b**) vibration frequency $f_1$. (**c**) initial phase $\phi_1$.

The high-frequency vibration signal in the form of (11) and $I = 2$ is considered. Besides the first vibration component mentioned above, the second vibration parameters

are set as $a_2 = 0.3$ mm, $f_2 = 20.0$ Hz, and $\varphi_2 = \pi/6$. In addition, the signal is also corrupted with Gaussian white noise. The SNR of the high-frequency vibration signal changes in 1-dB increments between $-2$ dB and 6 dB. For each SNR value, the above parameters estimation methods were performed to obtain the signal parameters, and one hundred simulations were performed.

Regarding different parameter-estimation methods, Figure 4 gives the measured NRM-SEs of the high-frequency vibration signal parameters in different SNR circumstances. Here, in the STFT-based rough estimation method, we also chose 20 and 30 as window widths. In the latter two methods, the optimal window width, $h_{op}$, is 26, which is determined by maximizing MLF in (23). Different window widths make in the STFT-based rough estimation method produce different estimation accuracies. It can be seen from the Figure 4 that the improved STFT-based fine estimation had the best performance, except for the initial phase, $\phi_2$. The proposed method is based on MLF to estimate the optimal vibration parameters. Since the amplitude of the second vibration error is relatively small, the MLF may be insensitive to the initial phase $\phi_2$ of the second vibration error. Therefore, when the optimal MLF value is obtained, there is still a great difference between the initial phase and the true value. In any case, even in rather low SNR, the proposed estimation method has better estimation accuracy and stability.

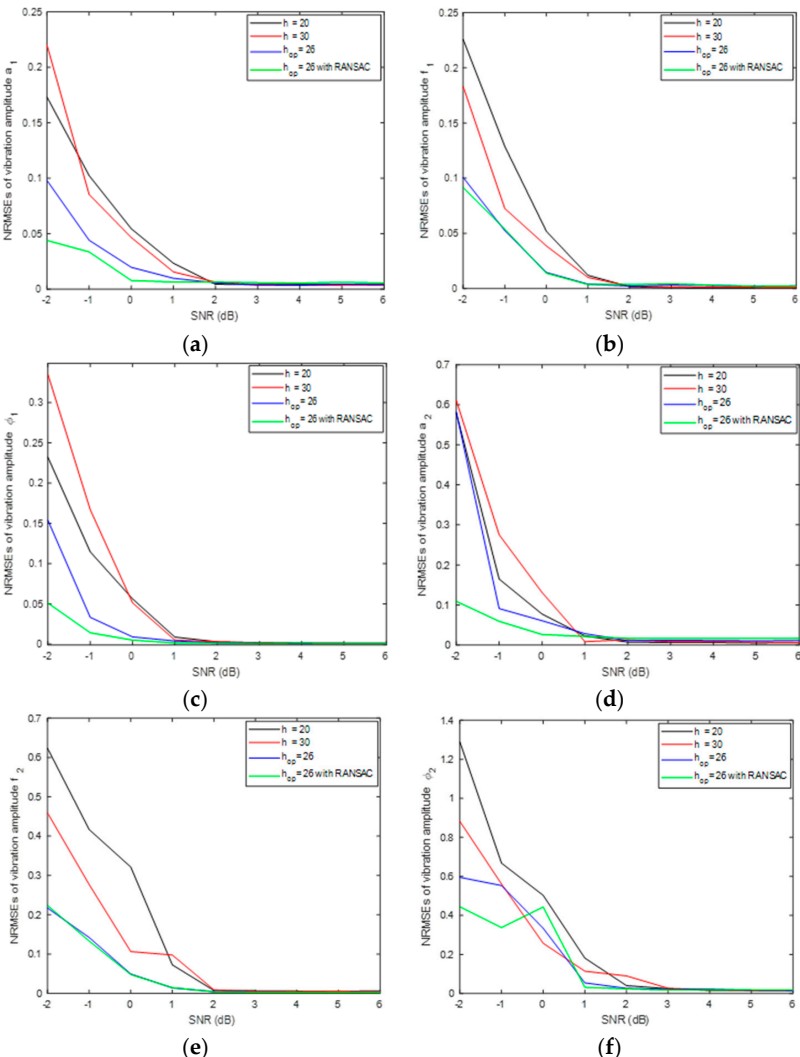

**Figure 4.** The estimation accuracy of signal parameters for different estimation methods. (**a**) vibration amplitude $a_1$; (**b**) vibration frequency $f_1$; (**c**) initial phase $\phi_1$; (**d**) vibration amplitude $a_2$; (**e**) vibration frequency $f_2$; and (**f**) initial phase $\phi_2$.

To sum up, from the perspective of parameter estimation, the proposed improved STFT-based fine estimation method showed better performance than that of the rough estimation method for both single-component and multi-component high-frequency vibration signals.

### 4.2. Single-Component Vibration Error Compensation

In this sub-section, simulations are given to evaluate the compensation performance of the proposed vibration error-compensation method for THz-SAR imaging. The THz-SAR system parameters are shown in Table 1. When the echo signal of multi-point targets is generated, single-component vibration error is added to it. To estimate and compensate the vibration error, the dominant scatter point is firstly selected according to the energies of range bins from the unfocused image. Then, after the target is moved to the image center by a circular shift, the SoI in range-compressed phase history domain is achieved through inverse FFT. To verify the outstanding performance of the proposed vibration error compensation method, the results obtained by the STFT-based rough estimation method and the STFT-based rough estimation method with the optimal window width are also given here, respectively.

**Table 1.** System parameters.

| Parameter | Value |
|---|---|
| carrier frequency | 220 GHz |
| signal bandwidth | 3 GHz |
| pulse width | 1.5 μs |
| platform velocity | 50 m/s |
| slant range | 2296 m |
| sample frequency | 2.8 GHz |
| pulse repetition frequency | 1050 Hz |

First, the IF of the SoI is estimated by the STFT and the vibration frequency is obtained from the amplitude spectrum of the IF. On the basis of the estimated vibration frequency, we can obtain the vibration amplitude and initial phase. Table 2 lists the estimated parameters and MLF values. Figure 5 shows the true vibration error and the estimated vibration error. We can see from the Table 2 that the estimated values by the STFT with $h = 20$ have a large deviation from the true values. Hence, the red line in Figure 5 shows that the estimated vibration displacement also deviates from the true one, especially at extreme points of vibration. Accordingly, the MLF value is much smaller than the true MLF value, also indicating that the performance of the method is poor. Although the performance of the STFT with the optimal window width $h_{op} = 26$ is improved, the estimates result still diverges from the true values, due to the singular IF estimates. Fortunately, however, Table 2 shows that the parameter-estimate values using the improved STFT-based fine estimation method are consistent with the true ones. It means that the improved method greatly suppresses both the effects of window width and the singular values of IF estimation. It is also verified by the MLF value in Table 2 and the estimated vibration displacement in the green line of Figure 5.

**Table 2.** Estimated parameter values.

| Parameter | $a_1$ (mm) | $f_1$ (Hz) | $\phi_1$ (rad) | G |
|---|---|---|---|---|
| true values | 2.00 | 1.00 | 1.05 | 511.39 |
| $h = 20$ | 2.30 | 9.96 | 0.85 | 119.31 |
| $h_{op} = 26$ | 2.12 | 9.98 | 1.07 | 430.24 |
| $h_{op} = 26$ with RANSAC | 2.02 | 9.98 | 1.05 | 495.60 |

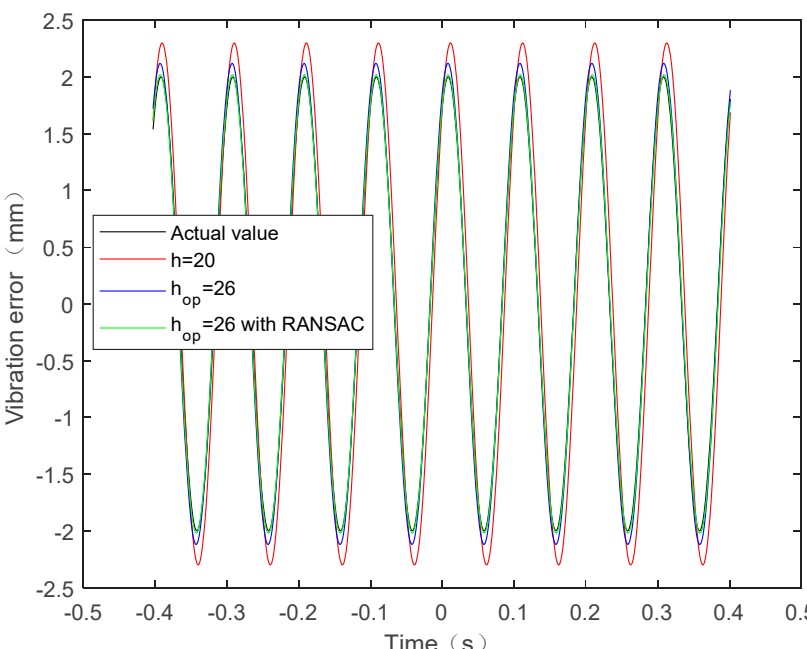

**Figure 5.** True and estimated single-component vibration error.

The vibration error is compensated for in the imaging process, and the performance of different estimation methods are then analyzed and compared. Figure 6a shows the imaging results of the RD algorithm without considering the vibration error. It is obvious that the image quality is seriously degraded due to the paired echoes caused by uncompensated vibration error. After the vibration error is estimated and compensated based on the above methods, Figure 6b–d gives the imaging results. The phase difference between the true vibration phase and the estimated vibration phase is given in Figure 7. We can see from Figure 6a,b that the image quality, after vibration error compensation, has improved, but it is still affected by paired echoes. This is caused by the residual period-modulated phase greater than $\pi/4$, which is shown in the red line of Figure 7. Meanwhile, the green line of Figure 7 shows that the residual phase is no greater than $\pi/4$, but is still periodically modulated. Therefore, the imaging quality of Figure 6c is further improved, but its side lobes are still relatively high. Fortunately, the remainder phase error is smaller than $\pi/4$ and no longer periodically modulated after compensation by the proposed method. Therefore, the paired echoes are perfectly suppressed, as shown in Figure 6d.

Finally, we performed a quantitative analysis to assess image quality. The center point target of Figure 6b–d is intercepted using a rectangular window. Figure 8 shows the resulting azimuth-dimensional amplitude profiles. Table 3 lists the impulse response width (IRW), peak sidelobe ratio (PSLR) and integral sidelobe ratios (ISLR) for all vibration compensation methods. It is clear from Figure 8 and Table 3 that the STFT-based rough estimation method with $h = 20$ is too poor to improve image quality. Correspondingly, after vibration error compensation by the STFT with $h_{op} = 26$, the image quality was improved, but its PSLR and ISLR were still high, as shown in red line of Figure 8 and Table 3. The great thing is that there are no longer paired echoes in the SAR images and the imaging quality indices are consistent with the theoretical ones after vibration error compensation by the STFT-based fine estimation method, which validate that the improved method can solve the problem of the former two methods and has the best performance.

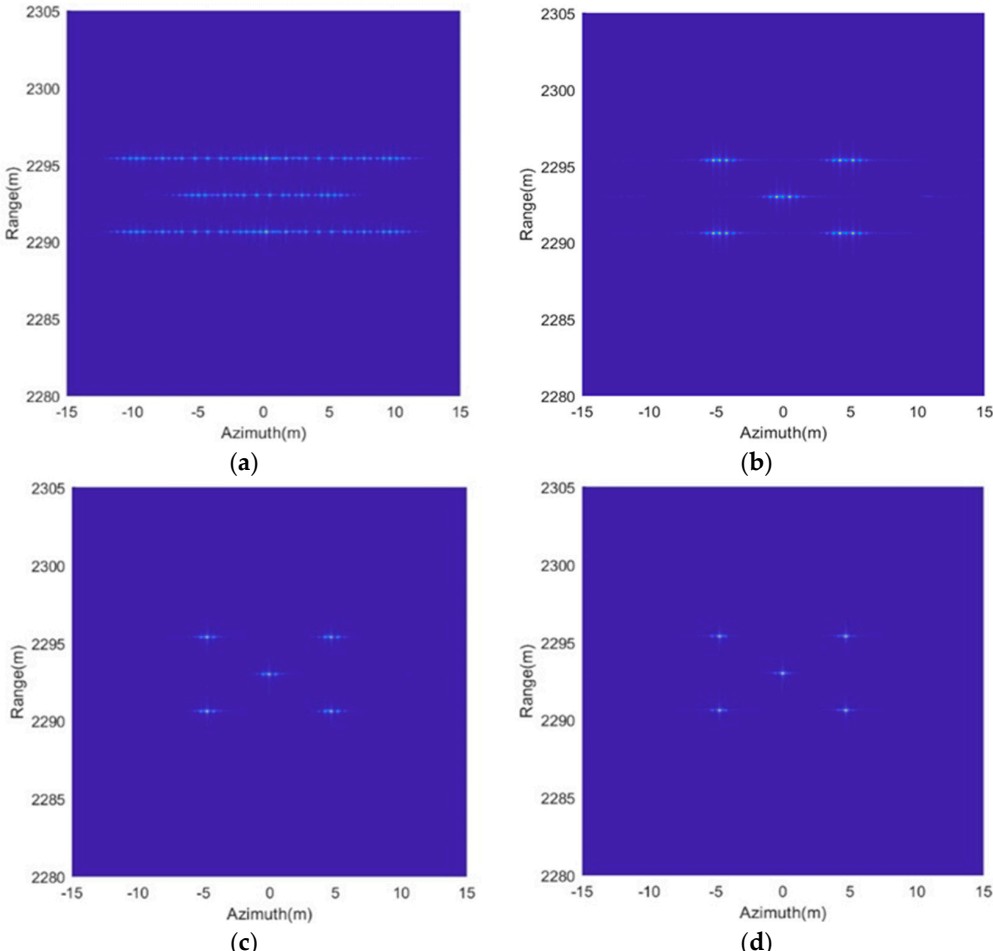

**Figure 6.** Comparison of compensation effect. (**a**) No compensation; (**b**) compensation based on STFT with $h = 20$; (**c**) compensation based on STFT with $h_{op} = 26$; (**d**) compensation based on STFT with $h_{op} = 26$ and RANSAC.

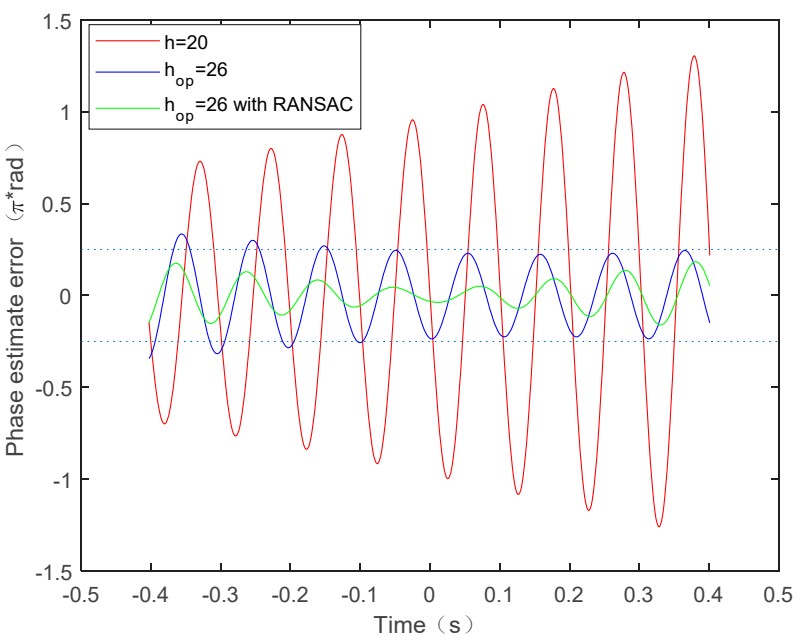

**Figure 7.** Phase estimate error.

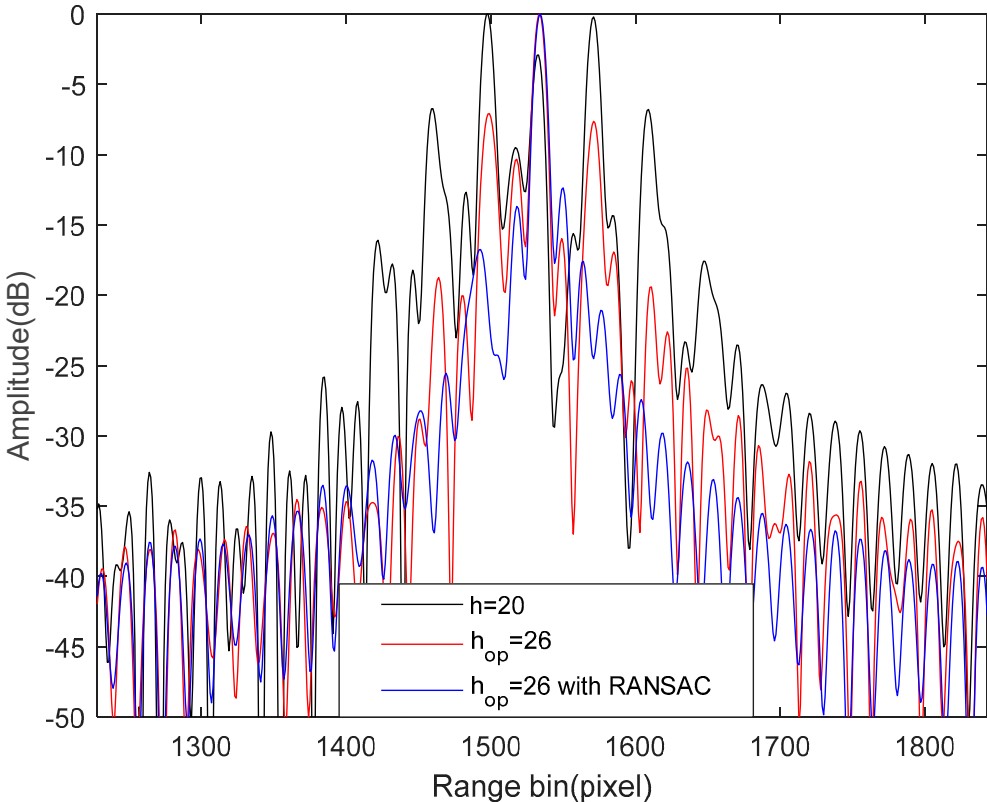

**Figure 8.** Azimuth-dimensional amplitude profiles of center-point target.

**Table 3.** Imaging quality evaluation after vibration error compensation.

| Parameter | IRW (m) | PSLR (dB) | ISLR (dB) |
|---|---|---|---|
| theoretical values | 0.050 | −11.95 | −13.73 |
| $h = 20$ | 0.064 | −0.21 | −5.38 |
| $h_{op} = 26$ | 0.053 | −7.10 | −8.57 |
| $h_{op} = 26$ with RANSAC | 0.051 | −12.37 | −11.48 |

*4.3. Multi-Component Vibration Error Compensation*

In this sub-section, the simulation results of multi-component vibration error compensation are provided to assess the performance of the proposed vibration error compensation method for THz-SAR imaging. Similarly, when the echo signal of multi-point targets is generated, the two components high-frequency vibration error is added. The SoI containing vibration error is extracted firstly according to the above steps. Similarly, the results obtained by the rough estimation method based on the STFT-based rough estimation method and the STFT-based rough estimation method with the optimal window width are also given here.

Using the STFT, we can obtain the IF of the SoI, and the vibration frequency is obtained from the amplitude spectrum of the IF. On the basis of the estimated vibration frequency, the vibration amplitude and initial phase are obtained through the LS regression. Table 4 lists all the estimated parameters and MLF values. Figure 9 shows the true vibration error and the estimated vibration error. It can be seen from the Table 4 that the estimated values by the STFT with $h = 20$ have a large deviation from the true ones. Hence, we can see from the red line of Figure 9 that the estimated vibration displacement also deviates from the true ones, especially at extreme point of vibration. Accordingly, the MLF value is much smaller than the true MLF value, implying that the performance of the STFT-based rough estimation method is poor. Although the performance of the STFT with the optimal window width $h_{op} = 26$ is improved, the estimates still diverge from the true values due to

the singular IF estimates. However, of value is that the estimated values of parameters by the proposed improved STFT-based method are consistent with the true ones, as shown in Table 4. It implies that the improved method greatly suppresses both the effects of window width and the singular values of IF estimation, which is also verified by the MLF value in Table 4 and the estimated vibration displacement in the green line of Figure 9.

**Table 4.** Estimated parameter values.

| Parameter | $a_1 \mid a_2$ (mm) | $f_1 \mid f_2$ (Hz) | $\phi_1 \mid \phi_2$ (rad) | G |
|---|---|---|---|---|
| true values | 2.00 \| 0.60 | 10.00 \| 20.00 | 1.05 \| 0.52 | 511.48 |
| $h = 20$ | 1.85 \| 0.66 | 9.96 \| 20.23 | 0.93 \| 0.62 | 172.14 |
| $h_{op} = 26$ | 1.90 \| 0.64 | 9.97 \| 19.94 | 1.09 \| 0.62 | 386.56 |
| $h_{op} = 26$ with RANSAC | 2.01 \| 0.59 | 9.97 \| 19.94 | 1.06 \| 0.56 | 464.39 |

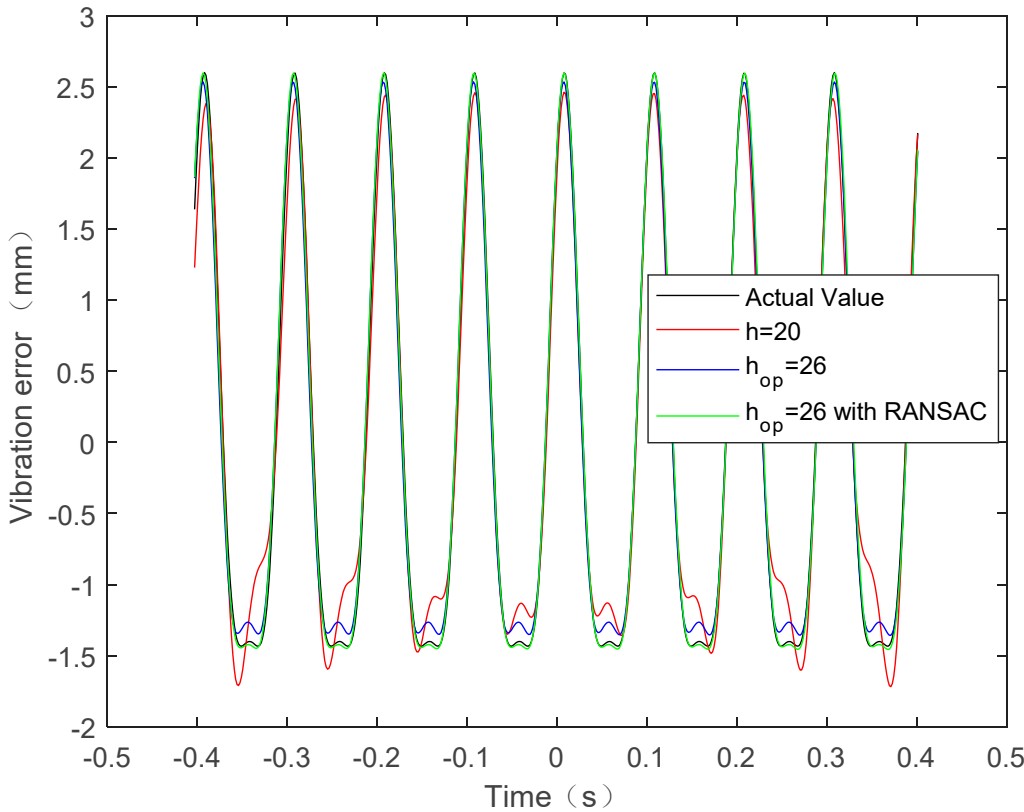

**Figure 9.** True and estimated multi-component vibration error.

The vibration error is compensated for in the THz-SAR imaging process, and the performance of different estimation methods are then analyzed and compared. Figure 10a shows the focusing results of the RD algorithm. Due to vibration error, the SAR image are completely defocused. Figure 10b–d shows the imaging results after the vibration error is compensated for based on the above methods, respectively. Figure 11 also gives the phase difference between the true vibration phase and the estimated vibration phase. We can see from Figure 10a,b that the image quality after vibration error compensation is improved, but it is still affected by paired echoes. This is caused by the residual period-modulated phase error greater than $\pi/4$, which is shown in the red line of Figure 11. Meanwhile, the green line of Figure 11 shows that the residual phase is no greater than $\pi/4$, but is still periodically modulated. Therefore, the imaging quality of Figure 10c is further improved, but its side lobes are still relatively high. Fortunately, the residual phase error is smaller than $\pi/4$ and no longer periodically modulated after vibration error compensation by the

proposed improved method. Therefore, the SAR image is fully focused, which is shown in Figure 10d.

Finally, we will make a quantitative analysis to assess the image quality. The center point target of Figure 10b–d is intercepted using a rectangular window. Figure 12 gives the resulting azimuth-dimensional amplitude profiles. Table 5 lists the imaging quality indices for all vibration compensation methods. We can see from Figure 12 and Table 5 that the STFT-based rough estimation method with $h = 20$ is too poor to improve the image quality. Correspondingly, after vibration error compensation by the STFT with $h_{op} = 26$, the image quality is improved, but its PSLR and ISLR are still high, as shown by red line of Figure 12 and Table 5. The value of the proposed method is demonstrated in that there are no longer paired echoes in the SAR images and the imaging quality indices are consistent with the theoretical ones after vibration error compensation by the STFT-based fine estimation method, validating that the improved method can solve the problem of the former two methods and has the best performance.

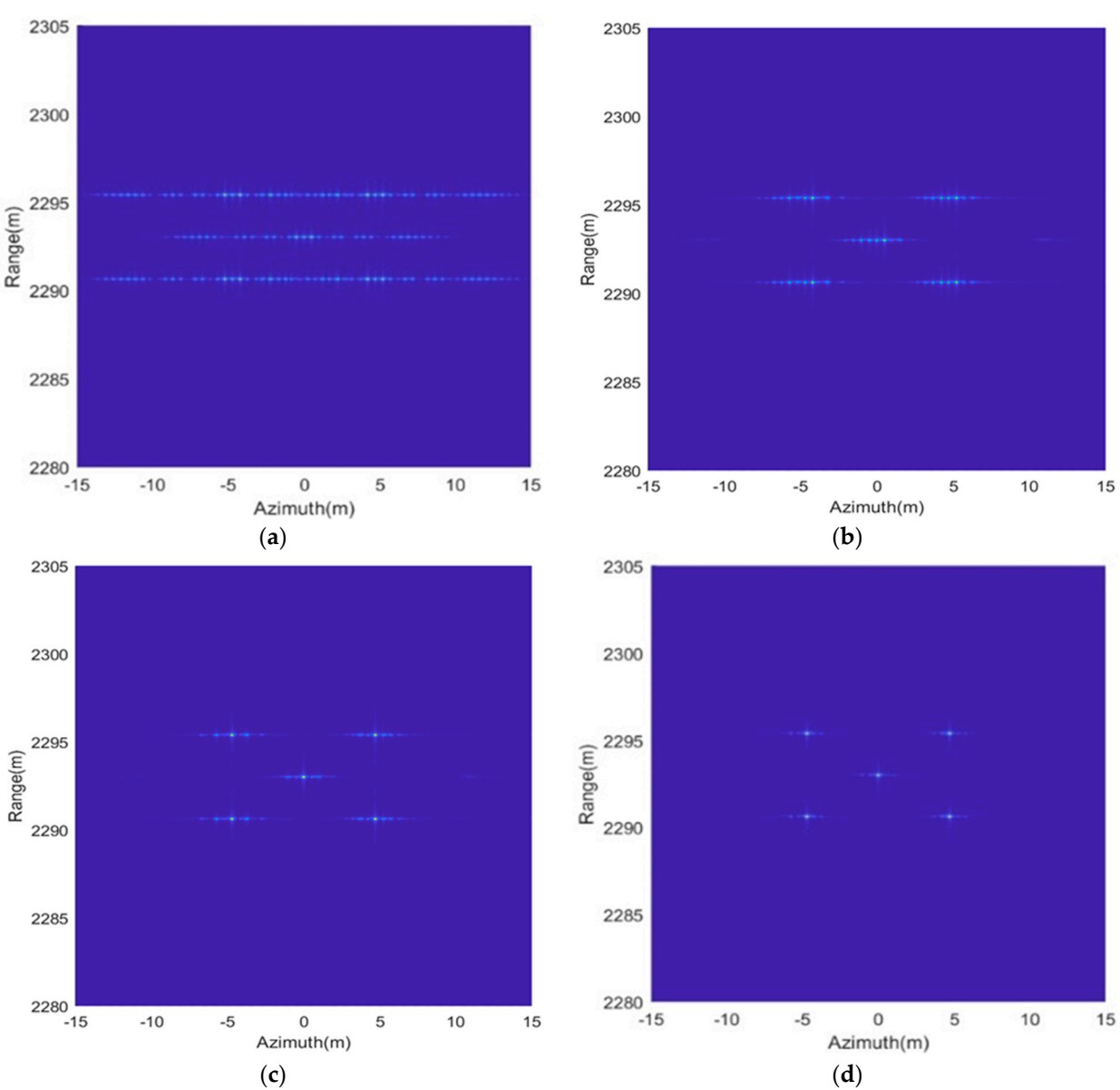

**Figure 10.** Comparison of compensation effect. (**a**) No compensation; (**b**) compensation based on STFT with $h = 20$; (**c**) compensation based on STFT with $h_{op} = 26$; (**d**) compensation based on STFT with $h_{op} = 26$ and RANSAC.

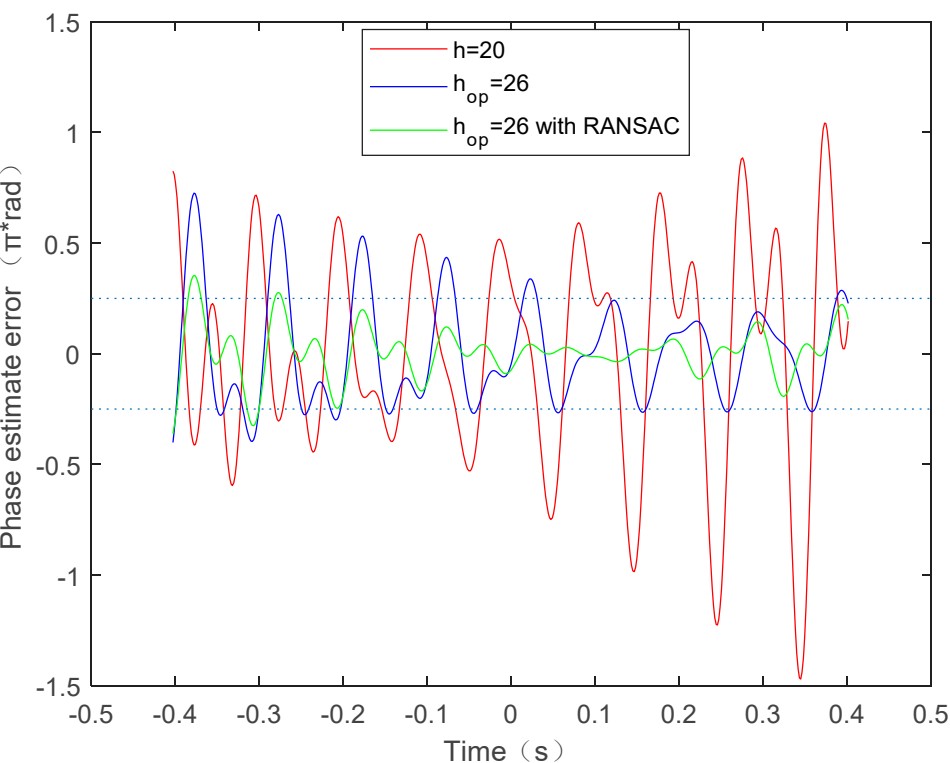

**Figure 11.** Phase estimate error.

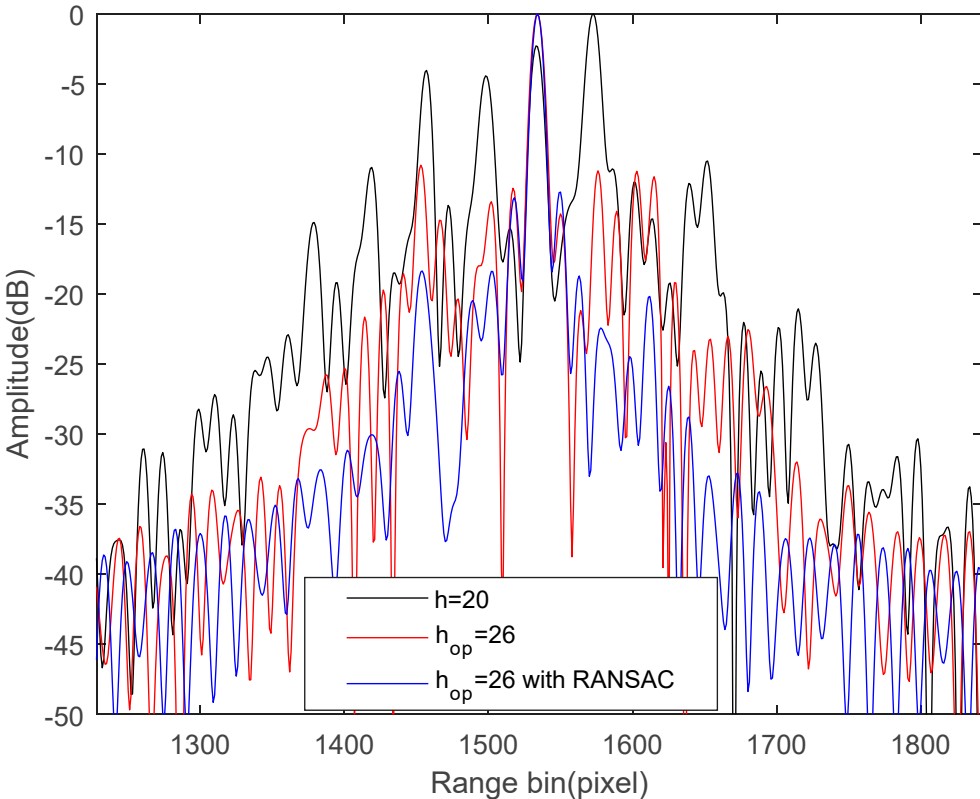

**Figure 12.** Quality evaluation diagram of multi-component compensation.

**Table 5.** Quality evaluation table of multi-component compensation.

| Parameter | IRW (m) | PSLR (dB) | ISLR (dB) |
|---|---|---|---|
| theoretical values | 0.050 | −11.95 | −13.73 |
| $h = 20$ | 0.074 | −2.29 | −5.76 |
| $h_{op} = 26$ | 0.071 | −11.20 | −10.03 |
| $h_{op} = 26$ with RANSAC | 0.062 | −12.72 | −10.09 |

## 5. Conclusions

To compensate for the non-negligible phase of periodic modulation induced by high-frequency vibration error of a radar platform, a novel vibration error compensation method based on STFT with the combination of the optimal window width and RANSAC was proposed in this paper. The method not only reduced the influence of the STFT window widths but also avoided the singular values of IF estimation in the LS regression. Therefore, regardless of whether the SNR is high or low, the proposed method can fulfill the focusing requirements of the image. Finally, our simulation results have verified its validity. However, at present, the low power of terahertz radiation sources means that such a radar must operate at a close range, and there are few THz-SAR systems installed on aircraft for the time being. Therefore, we will use the measured SAR data to further test the effectiveness of the algorithm in the future. Meanwhile, all the existing high-frequency vibration error estimation methods ignore the effect of low-frequency motion errors. Thus, research on the synchronous estimation of low-frequency motion error and high-frequency vibration error in THZ-SAR imaging is the key to work advancing this technology in the future.

**Author Contributions:** Conceptualization, Y.L.; Funding acquisition, X.D. and Y.Z.; Methodology, Y.L.; Project administration, X.D. and Y.Z.; Resources, Q.Z.; Software, Q.W.; Validation, J.J. and Q.Z.; Writing—original draft, Q.W. and J.J.; Writing—review and editing, Y.L., Q.Z. and Y.Z. All authors have read and agreed to the published version of the manuscript.

**Funding:** This research was supported in part by the Natural Science Foundation of Shanghai (Grant No. 21ZR1444300), in part by the National key R and D Project of China (Grant No. 2018YFF01013003), in part by the National Natural Science Foundation of China (Grant No. 61722111, 61731020, 42005110), the 111 Project (Grant No. D18014), the International Joint Lab Project of Science and Technology Commission Shanghai Municipality (Grant No. 17590750300), the Key Project of Science and Technology Commission Shanghai Municipality (Grant No. YDZX20193100004960).

**Institutional Review Board Statement:** Not applicable.

**Informed Consent Statement:** Not applicable.

**Conflicts of Interest:** The authors declare no conflict of interest.

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
