# Peer review of "A High-Frequency Vibration Error Compensation Method for Terahertz SAR Imaging Based on Short-Time Fourier Transform"

_applsci, doi:10.3390/app112210862_

Round 1

Reviewer 1 Report

A high frequency vibration error compensation method for terahertz SAR imaging based on short-time Fourier transform

In the manuscript authors proposed method to estimate vibration error. The novelty of the manuscript lies in the fact that a modified error estimation scheme is proposed. The manuscript is written on a relevant topic, however, it is often difficult to understand what the authors mean. It seems to me that the manuscript needs editing in English. In general, the study is applied. I recommend the manuscript for additional review to specialists in THz image quality.

Introduction. I think that section Introduction can be improved. I recommend to authors describe the reasons of the errors in details. Authors may clarify the range of error changes. I believe the estimation of these errors is very important and may be relevant for astronomical mm/submm telescopes.

As the method is based on the simulation I recommend expand the conclusions and discuss the method, its advantages numerical data.

Minor notes:

line 37. «...infrared wave [1,2]». Maybe you mean between microwave and infrared waveS [1,2].

Reviewer 2 Report

This paper reports a method to improve the quality of terahertz SAR images. The authors applied MLF and RANSAC to the STFT-based estimation in order to obtain higher accuracy. 

The introduction details the background and fundamentals of the methods enough, and the results are clearly presented. The results indicate that the proposed procedure successfully reduces the influence of high-frequency vibration of the platform. 

There are several points to be explained or fixed before publication:

(1) lines 223-224 and Eq.25 are difficult to understand. A sufficient description or reference is necessary. 

(2) The resolution of Fig.2 is too low.

(3) Readers may want to know why "the improved STFT-based fine estimation has (the) best performance, except for initial phase 305 ?2." (lines 305-306)

(4) In Table 3, PSLR of "h=20" is -3.37 dB. In Fig. 8, the central peak is lower than the side peaks. So, how do you define and estimate this value?

(5) Similarly to (4), Table 5 should be considered again. 

(6) In Table 5, the PSLR of "hop=26" is -6.51 dB. It seems different from the data presented in Fig 12.

Other minor points are checked in the attached PDF file.
